# The Impact of Quality of Work Organization on Distress and Absenteeism among Healthcare Workers

**DOI:** 10.3390/ijerph192013458

**Published:** 2022-10-18

**Authors:** Nicola Magnavita, Carlo Chiorri, Leila Karimi, Maria Karanika-Murray

**Affiliations:** 1Postgraduate School of Occupational Health, Università Cattolica del Sacro Cuore, 00168 Rome, Italy; 2Department of Woman, Child & Public Health Sciences, Fondazione A. Gemelli IRCCS, 00168 Rome, Italy; 3Department of Educational Sciences, University of Genova, 16126 Genova, Italy; 4School of Applied Health, Psychology Department, RMIT University, Melbourne, VIC 3000, Australia; 5School of Medicine and Healthcare Management, Caucasus University, Tbilisi 0141, Georgia; 6Department of Psychology, Nottingham Trent University, 50 Shakespeare Street, Nottingham NG1 4FQ, UK

**Keywords:** occupational stress, effort/reward imbalance, demand, control, social support, job strain, overcommitment, musculoskeletal disorders, mental health, sickness absence

## Abstract

The quality of work organization may be responsible not only for reduced productivity but also for an increased risk of mental and physical disorders. This study was aimed at testing this hypothesis. Workers of a local health unit in Italy were asked to fill out the Work Organization Assessment Questionnaire (WOAQ) during their periodic medical examinations in the second half of 2018. On the same occasion, they also completed the Demand/Control/Support (DCS) measure of job strain, the Effort/Reward Imbalance (ERI) questionnaire, and the General Health Questionnaire (GHQ12) to assess psychological health. A total of 345 workers (85.8%) completed the survey. Linear regression analysis showed that the quality of work organization was inversely proportional to psychological health problems (*p* < 0.001). Occupational stress, measured both by job strain and ERI, was a moderating factor in this relationship. The relationship between the WOAQ and psychological health, moderated by job strain or ERI, remained highly significant even after adjustment for sex, age, social support, and overcommitment. Regression models explained over 40% of the shared variance of the association between quality of work organization and psychological health. The quality of work organization significantly predicted the risk of sickness absence for musculoskeletal disorders (OR = 0.984, CI95% 0.972–0.996) and for other health problems (OR = 0.977, CI95% 0.967–0.988). A continuous improvement of work organization must consider not only the clients’ or production needs but also the well-being of workers.

## 1. Introduction

The quality and organization of the work in public health and healthcare organizations is fundamental to maximizing the effectiveness of health services, especially under restricted resources. In Italy, the universal health coverage provided by the national health system has allowed significant improvements in public health, even with a decrease of resources [1]. However, since each region is responsible for its own health services planning and delivery, wide disparities in performance exist between regional health services [2]. Local healthcare organizations are therefore challenged to deliver high-quality care with a gradual decrease in public health expenditure. Other countries worldwide are in a similar situation.

A great variety of concepts and tools have been used to solve labor problems in healthcare. Industrial improvement approaches such as benchmarking [3] or lean management are increasingly being adopted [4,5]. However, healthcare environments can be complex, limiting the ability to obtain optimal layout solutions [6]. Less-than-satisfactory outcomes have often been achieved through implementation of industrial and business methods in healthcare [7], and this has been attributed to the special characteristics of healthcare, which cannot be based solely on economic criteria [8]. As things stand, healthcare workers (HCWs) often witness conflicting views by national, regional, and local managers on the most appropriate policies and strategies to be implemented in order to increase efficiency.

Moreover, healthcare facilities are organized in a way that sees modern management based on the care complexity of patients and the intensity of the care that they need [9]. This contrasts with the traditional system, which is based on operating units that trace the different specialties and on districts that follow the territorial distribution of services. In such conflicting situations within the same company, individual employees may find themselves working in very different organizational settings. The organizational climate has long been recognized as one of the risk factors for workers’ health [10,11,12]. Comprehensive literature reviews showed that work environment factors highly correlate to employees’ health and well-being [13] and to quality of life and quality of working life among healthcare workers [14]. There is strong evidence that low organizational justice, high job strain, high effort/reward imbalance, and low social support may increase the risk for musculoskeletal disorders [15], and that poor organization of work may be responsible for the emergence of mental and physical health problems [16], particularly in hospitals, due to the concurrence of workload, psychosocial, and ergonomic factors [17]. On the contrary, improved quality and organization of work that allows implementing multi-domain interventions (i.e., including healthcare provision, service coordination, or work accommodation components) can help to increase job retention and promote the return to work of people with mental and musculoskeletal health problems [18,19]. There is strong evidence that quality of work (the work environment and way that work is organized) impacts psychological health [20], reducing psychological distress and increasing well-being in workers [21].

The quality of work organization can be evaluated through two different approaches. The first consists of measuring the organization’s impact on workers’ psychosocial outcomes. For example, organizational justice [22,23] is significantly associated with the mental well-being of workers and with absence due to lower-back pain [24]. The second approach is to directly ask workers how they experience specific work organization factors. This approach gave rise to the Work Organization Assessment Questionnaire [25], which has been applied in public sector workers [26] and in the healthcare sector [27,28,29] in European and non-European countries.

There is strong evidence, from meta-analytic research, of the effects of the psychosocial work environment on mental health [30]. Those studies supported expectations about similar effects among the quality of work organization and psychological health outcomes, and the role of perceived stress in this relationship. Indeed, the psychosocial work environment, in interaction with genetic and epigenetic factors [31], contributes to psychological outcomes, such as stress-related disorders [32], and behavioral outcomes, such as sickness absenteeism [33]. This is especially pronounced in emotional labor (i.e., all service jobs that cannot be described only by the physical component but must consider the interaction of users and their emotions), which is common in healthcare [34]. The literature has strongly supported the moderating role of emotion in this relationship [35]. The performance of emotional labor can have both negative and positive consequences for workers, depending on the different forms of emotion management involved [35]. Workers’ perceptions of stress can be measured in two complementary dimensions: job strain [36] and effort/reward imbalance [37]. In previous studies, the demand/control and effort/reward imbalance models independently predicted poor self-reported health status; however, combining both models was a better predictor of self-reported health status and any chronic condition than either model alone [38,39]. Systematic reviews demonstrated that low organizational justice may be involved in developing adverse health and occupational outcomes in HCWs [40]. Prospective studies indicated with moderate evidence that low organizational justice is associated with a greater risk of developing common mental health problems [41]. Unfavorable work conditions are associated with burnout in nursing [42]. When examining these relationships, it is important to consider overcommitment and social support because these factors are also strong independent predictors of psychological health [43].

In this study, we aimed to verify the assumption that the organization of work is associated with workers’ mental health status. Furthermore, we expected that perceptions of occupational stress would moderate this relationship. Finally, we studied the relationship between work organization and sickness absence due to musculoskeletal disorders or other diseases. We tested the following hypotheses:

**H1:** 
*Quality of work is positively associated with psychological health;*


**H2:** 
*The relationship between quality of work and psychological health is moderated by job strain;*


**H3:** 
*The relationship between quality of work and psychological health is moderated by effort/reward imbalance;*


**H4:** 
*Quality of work is negatively associated with sickness absenteeism (for both musculoskeletal and other health problems).*


## 2. Materials and Methods

### 2.1. Design

The study was cross-sectional and descriptive. We followed the STROBE guidelines [44].

### 2.2. Population

European directives state that workers exposed to occupational risk will be assessed by an occupational physician who will verify their suitability for work. All workers of the Local Health Unit Roma 4, Civitavecchia, in the Latium region of Italy, who were called for the periodic visit in the second half of 2018 were asked to complete a questionnaire containing measures of their work experience and psychological health as well as demographic and job-related information (sex, age, and type of job). Age was measured in years. Job type was measured as three professional categories: physician, nurse, and support staff. The workers invited to participate were all those who: (1) had been classified as “exposed to occupational risk” by the employer; (2) had undergone a preventive medical examination at least a year earlier; (3) had to undergo a new examination for the expiration of the judgment of suitability at risk. Participation in the survey was voluntary.

### 2.3. Questionnaire

The Italian version [45] of the Work Organization Assessment Questionnaire (WOAQ) [25] was used to evaluate the overall quality of the work environment and the way that work was organized. The questionnaire comprises 28 items, grouped into five categories: (1) quality of relationships with management, (2) reward and recognition, (3) workload issues, (4) quality of relationships with colleagues, and (5) quality of physical environment. Using their knowledge and experience, participants were asked to evaluate each aspect of their work in terms of how problematic (or good) it had been over the last six months using a five-point Likert-type scale (5 = very good to 1 = major problem). The WOAQ uses situational rather than psychological reasoning; it asks, “How good or poor do you and your colleagues think this aspect of work design (or management) is?” rather than “To what extent are you upset or distressed by this aspect of work design (or management)?”. The work organization assessment index gives a score from 28 to 140. Higher scores indicate higher quality of work organization. Previous studies showed that global measure of WOAQ could be used as a single factor to assess the overall quality of work organization [27,28]. In this study, the reliability of the WOAQ was very high for the whole population (Cronbach’s alpha = 0.949) as well as each subgroup (Physician group Cronbach’s alpha = 0.949, Nurse group Cronbach’s alpha = 0.951, Support staff Cronbach’s alpha = 0.948).

The participants were also asked to fill out a questionnaire on job strain: the Italian version [46] of the Demand/Control/Support (DCS) questionnaire based on the Karasek model [36,47,48]. The DCS model postulates that job stress emerges from an interplay of job demands (e.g., quantitative workload, degree of difficulty, time available to perform tasks, role conflict, etc.) and job control (or decision latitude, e.g., the ability to make decisions about how to complete job tasks). Social support at work (i.e., the need to relate to others and to seek out help in accomplishing difficult tasks) was later included in the model when its moderating effect on job strain [49,50] and psychological health [51,52] was established. The DCS questionnaire comprises 17 questions to be rated on a 4-point Likert-type scale and provides scores of the demand (5 items, score from 5 to 20) and control (6 questions, score from 6 to 24) scales, whose weighted ratio, called “job strain” (JS), expresses perceived stress. The support scale measures social support with 6 questions (score from 6 to 24). In this study, the Cronbach’s alpha for demand was 0.737 (acceptable), for control 0.612 (acceptable), and for support 0.862 (very good).

The participants also were requested to fill in the Effort/Reward Imbalance (ERI) questionnaire by Siegrist [37,46,53,54]. The Siegrist’s ERI model takes into account the reward rather than the control structure of work, suggesting that psychological issues such as distress are due to a high degree of effort that is not adequately rewarded (e.g., in the form of pay, recognition, status, or career opportunities). This model is completed by a third, intrinsic component: overcommitment, which is defined as a personal set of attitudes, behaviors, and emotions reflecting excessive motivational striving combined with a strong desire for approval [55]. The ERI questionnaire comprises 23 questions to be rated on a 4-point Likert-type scale and measures the subjective effort made to work (effort, 6 questions, score from 6 to 24) and the material or intangible rewards received for the work completed (reward, 11 questions, score 11 to 44). The weighted ratio between effort and reward indicated the imbalance (ERI) that is the extrinsic component of stress. In addition, the model also included an intrinsic component of work stress: the “overcommitment” (OC) coping pattern, which has an independent role in explaining the psychological health of workers in many studies [56]. The overcommitment scale is composed of 6 questions, to be rated on a 4-point Likert-type scale with scores ranging from 6 to 24. In this research, the Cronbach’s alpha of the subscales was 0.852 for effort, 0.855 for reward, and 0.861 for overcommitment (very good).

Psychological health was measured using the Italian version [57] of Goldberg’s General Health Questionnaire (GHQ12) [58]. The questionnaire presents 12 questions answered on a 4-point Likert-type scale. The overall score values ranged from 12 to 48, with a higher score indicating a higher level of mental discomfort. Cronbach’s alpha was 0.899, suggesting an optimal internal consistency of the scale in this sample.

The participants were also asked to indicate with a binary response (yes/no) whether in the year prior to the visit, they had abstained from work due to musculoskeletal problems, and if they had abstained for other diseases.

### 2.4. Ethics

All the workers signed consent to the anonymous processing of their personal data and publication of the results from the analyses, which included their consent related to the present study. The study was conducted in accordance with the Declaration of Helsinki [59] and approved by the Ethics Committee of the Università Cattolica del Sacro Cuore, Rome, Italy (protocol code 1226, 24 November 2016).

### 2.5. Statistics

The distribution of scores was analyzed by measuring central tendency (mean, median, mode) and dispersion (standard deviation). We did not need to normalize the data before the statistical analyzes because, as reported by Lumley et al. [60], the assumption of normality is only required for small samples due to the central limit theorem. With sample sizes exceeding 30, as was the case here, violations of the normality assumptions are not problematic, and they become less and less problematic, even when extreme, as the sample size increases.

Comparisons between the different professional categories, sexes, and age groups of workers were carried out using one-way ANOVAs, followed by Bonferroni’s post-hoc tests. The relations between these variables were examined using Pearson correlation.

Once the presence of a significant correlation between the variables was found, we specified a linear regression model that included interactions in order to test moderation effects. We hypothesized that the work organization could act as a predictor of psychological health (as measured by the GHQ12), with stress as a moderating factor in this relationship. Social support, overcommitment, age, and sex were considered confounding factors.

Logistic regression analysis was used to evaluate the association between WOAQ or stress variables and the number of days of absence due to musculoskeletal disorders or other diseases in the year preceding the visit. The estimated effect was presented in terms of odds ratio (OR) and 95% confidence intervals (95%CI). Each of the variables was initially posited as a predictor in univariate models in which the absence of back pain or for other musculoskeletal problems was the response variable. Subsequently, multivariate logistic regression models, adjusted for gender and age, were constructed.

The statistical analyses were carried out using SPSS 26.0 (IBM, Armonk, NY, USA) integrated by PROCESS version 4.1 [61].

## 3. Results

Overall, 345 workers of the 402 who had been invited agreed to participate in the survey (85.8% participation rate). The main reason for non-participation was the lack of time to complete the questionnaire. The sample was mainly composed of females (230, 66.7%) and nurses (201, 58.3%) among the professional categories. The average age was 43.44 years (s.d. 8.71). The characteristics of the sample corresponded to those of the employees of the Italian National Health Service [62]. See Table 1 for the participant demographics.

### 3.1. Intergroup Comparison

The comparison between professional categories (Table 2) indicated that the evaluation of work organization does not depend on the type of professional role. Physicians reported significantly greater control than other professional categories and greater effort than support staff, but the levels of stress and psychological well-being did not differ among the different categories. In comparing the different age groups, no differences were found in the evaluation of work organization, stress, or psychological health. More females reported that they were overcommitted and had a lower level of psychological health compared to males.

### 3.2. Relationships between Work Organization, Stress, and Psychological Health

Work organization and the variables that measure occupational stress and psychological health were significantly correlated (Table 3). The quality of work organization was inversely related to stress and low psychological health. Social support was inversely related to low psychological health while overcommitment was positively correlated with low psychological health.

Moderation analysis revealed that both the direct and indirect (via job strain, Table 4, or via ERI, Table 5) associations between work organization quality and psychological health were statistically significant after adjustment for gender, age, social support, and overcommitment. The conditional effects of the independent variable WOAQ at values of the moderator job strain were significant (*p* ≤ 0.05) at levels −0.3008, 0.0001, and 0.3008 (−SD, M, +SD). Similarly, the conditional effects of the independent variable WOAQ at values of the moderator ERI were significant (*p* ≤ 0.05) at levels −0.3827, 0.0001, and 0.3827 (−SD, M, +SD). Figure 1, Figure 2, Figure 3, Figure 4, Figure 5 and Figure 6 show the relationship between work organization quality and psychological health moderated by job strain and by effort/reward imbalance.

### 3.3. Relationship between Work Organization and Sickness Absence

During the year preceding the visit, 99 workers (29.2%) had been on sick leave for back pain or other musculoskeletal disorders and 150 workers (44.2%) had been on sickness absence for other causes.

Logistic regression was used to examine how work organization was related to absenteeism due to musculoskeletal disorders. In univariate analyses, work organization was significantly and positively associated with the occurrence of sickness absence separately for musculoskeletal disorders (Table 6) and other causes (Table 7). After addition of the control variables in the prediction model, the relationship between WOAQ and musculoskeletal disorders remained highly significant. Overcommitment was significantly associated with absenteeism for musculoskeletal disorders, but the relationship became non-significant when the demographic variables were included (Table 6). Stress (measured as an imbalance between effort and rewards) was positively correlated with sickness absence for other-than-musculoskeletal disorders, and social support was negatively associated with these problems. Both these relationships were stable after the addition of the demographic variables (Table 7).

## 4. Discussion

This study was conducted on a group of healthcare workers in a local health unit in Italy. The findings showed that quality of work organization was significantly associated, in a protective way, with psychological health. Specifically, occupational stress significantly moderated the relationship between work organization and psychological health. Work organization was also negatively associated with sickness absence, both from musculoskeletal problems and other causes. The hypotheses that motivated this study have been confirmed.

These results are in line with the literature [25,26,27,28,29] that the WOAQ appears to be a great instrument for assessing risk factors associated with employee health and health-related behavior. Improvements in work organization were associated with a higher level of organizational justice perceived by workers [64]. Perceived job characteristics and organizational justice could improve nursing care quality [65]. As mentioned earlier, evidence showed that low organizational justice is linked to adverse health and occupational outcomes in HCWs [40] and a greater risk of developing common mental health problems [41], whilst negatively experienced work conditions are linked with burnout in healthcare workers [42]. In reverse, however, effective work organization could help to improve workers’ mental health, reduce burnout and increase work engagement through meritocratic person-centered work culture, minimization of bureaucracy, and opportunities for employee professional development and self-care [66].

The WOAQ has been shown, here and elsewhere, to be a valid and functional measure of the organization of work. The WOAQ asks workers to express the opinion that they and their colleagues share of the organization of work. Objectivity, in this sense, was supported by the lack of significant differences in judgment between the various professional categories, age groups, and sexes. On the contrary, when workers were questioned about their perception of organizational justice, that is, the level of correctness of procedures, distribution of tasks and rewards, information flows, and relationships in the workplace, significant differences were observed between the various subgroups of HCWs, with doctors who complained of poor distributive justice, women who reported poor interpersonal justice, and elderly workers who reported less informative justice than the other subgroups of the healthcare population [24]. This difference in the distribution of responses indicates that the evaluation of work organization through a questionnaire such as the WOAQ is more objective than that based on the perception of organizational justice, in which the personal component of judgment has predominant importance. Another substantial difference between the assessment conducted with the WOAQ and that obtainable through the organizational justice questionnaires is that the former considers both ergonomic factors and human factors while the latter examines only human factors.

In the literature, the organization of healthcare work has also been studied with qualitative methods that demonstrate how staff is able to correctly describe problems and provide useful elements to management to improve a given situation [67]. However, these methods appear particularly suitable for small samples and have limited reproducibility. The use of a standardized questionnaire is recommended when considering large populations or when you want to check the effectiveness of the organization’s improvements over time. The WOAQ appears to be a valid and reliable tool for asking workers to evaluate the organization of their workplace.

Healthcare workplaces are complex and involve many critical elements. A systematic review and meta-analysis found that occupational exposure to ergonomic risk factors is highly prevalent in healthcare [68]. Other widespread hazards are shift work and violence. Good organizational climate, with increased workplace cohesion and involvement and decreased work pressure, may mitigate the negative health outcomes of shift work [69] and of workplace violence [70]. Social support is a known mediator in the relationship between organizational justice and health [71] and proved to reduce disability from all causes, such as depression and musculoskeletal diseases, in public sector employees [72]. Our findings confirmed the protective effect of social support and indicate that ERI may, conversely, increase the risk. In this regard, the Finnish Public Sector Study has shown that the association of low organizational justice and ERI determines a doubled risk of disability pension for musculoskeletal or depressive disorders, and that the highest risk of disability can be observed among those with work stressor combinations strain+ERI or strain+ERI+injustice, rather than those with single stressors [73]. As for overcommitment, which in our sample showed a weak association with musculoskeletal sickness before adjustment for age and sex, the results are in agreement with the literature review that expressed inconclusive evidence of the role of overcommitment and of its interaction with effort/reward imbalance in musculoskeletal disorders of healthcare workers [74].

The two stress models we used have different meanings and, consequently, different relationships with the outcomes. In the vast complex of care activities, each of the two stress models captures some specific aspects: the DCS more linked to physical loads and the ERI to intangible contents. In our sample, which brought together the many activities of a local healthcare unit, the variables deriving from the two models had modest associations with sickness absence. This result is in agreement with what was found in the literature. A German study showed that organizational justice was a significant independent predictor of musculoskeletal pain only among white-collar workers, whereas job strain had additive predictive utility exclusively among blue-collar workers; in reverse, ERI influenced pain-symptom reporting in both occupational groups [75]. In a heterogeneous population of such a local healthcare unit, the assessment of work organization quality with the WOAQ is a more effective predictor of sickness absences than measurement of occupational stress.

The identification of problems in the organization of work is the basis for the development of workplace health interventions and participatory ergonomics programs. These programs have been applied in different contexts, with encouraging results [76,77]. It is, however, evident that programs can only work with the concrete support of company management [78]. In the hospital, they were used not only to solve problems related to the manual handling of loads [79,80] or incorrect postures [81,82] but also to improve the organization of shifts [83], prevent workplace violence [84,85], control alcohol and drug use [86], and deal with relationship problems or multiple organizational issues [87]. Participatory simulation activities, with subsequent transferring and integrating of the simulation outcomes into the design of workplaces, have been proposed for improving hospital design projects [88]. In the company where this study was carried out, the experience of the participatory ergonomics groups (gruppi di ergonomia partecipativa/GEP^©^) was initiated [89]. Through this method, the staff of the various operating units were called to analyze the organization of work, to identify the critical points, and to propose and elaborate upon improvements by favoring the characteristics of cheapness, feasibility, and acceptability of the measures. Indeed, the role of participatory groups that can lead the implementation of interventions has been shown to be essential for the success of such programs [90]. The proposals shared by the workers were sent to company management, which evaluated their adoption. This method received an award in 2008 from the Italian Society of Ergonomics and was included among the best practices in the European campaigns “Lighten the load” in 2007 and “Safer and healthier work at any age—occupational safety and health in the context of an aging workforce” in 2016/17. We are confident that the results of this study can stimulate the development of participatory initiatives for the improvement of the organization in other healthcare companies.

The findings of this research have to be appreciated in light of its limitations. First, the cross-sectional nature of the study prevented us from inferring causality. However, it seems plausible to believe that work organization is a determinant of distress and to believe that reverse causality is unlikely. Second, the survey was carried out in a single health unit, and this requires the utmost caution in extrapolating the results to other work situations. Finally, the sample size was smaller in physicians and support staff compared to nursing. However, it is sensible to observe that there are no great differences between the various health units, and we can therefore believe that by applying the same method in other situations, comparable results can be obtained.

## 5. Conclusions

The quality of work organization, as measured by the WOAQ, has a significant impact on psychological health and sickness absences, supporting the importance and the need to continuously improve the organization of work to improve workers’ well-being, especially in the demanding healthcare sector.

## Figures and Tables

**Figure 1 ijerph-19-13458-f001:**
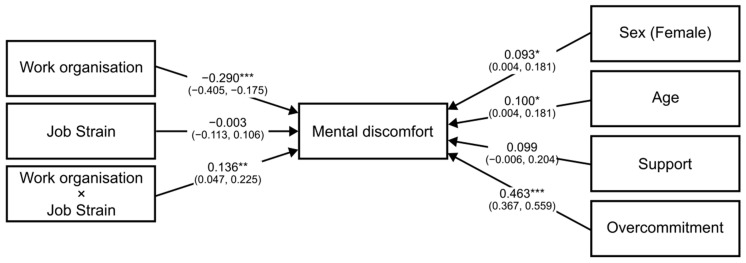
Diagram of the job strain model. Note: ***: *p* < 0.001; **: *p* < 0.01; *: *p* < 0.05.

**Figure 2 ijerph-19-13458-f002:**
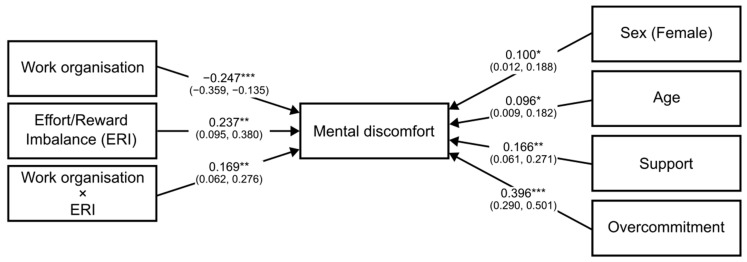
Diagram of the ERI model. Note: ***: *p* < 0.001; **: *p* < 0.01; *: *p* < 0.05.

**Figure 3 ijerph-19-13458-f003:**
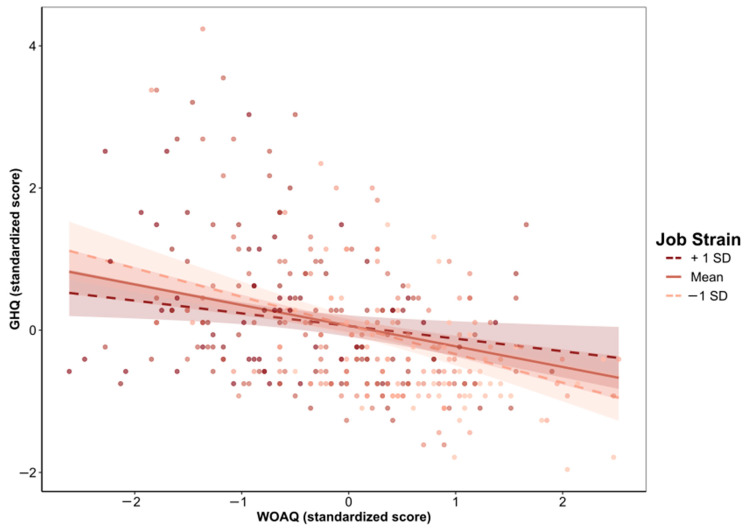
Simple slopes plot of the job strain model.

**Figure 4 ijerph-19-13458-f004:**
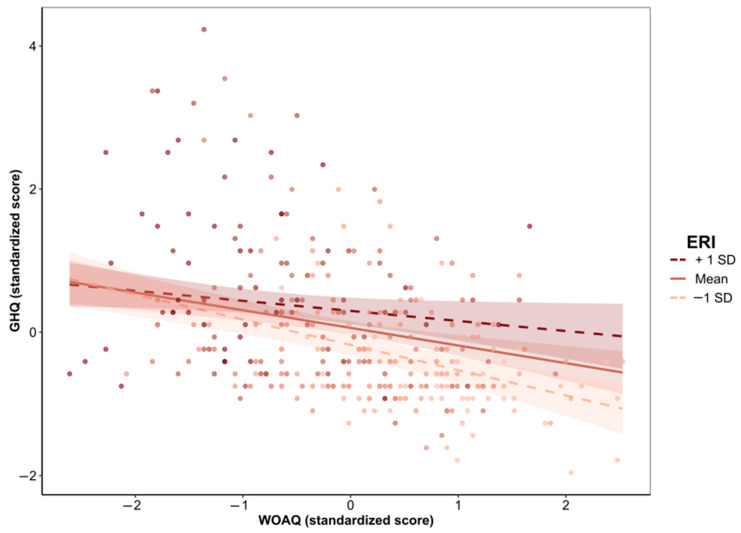
Simple slopes plot for the ERI model.

**Figure 5 ijerph-19-13458-f005:**
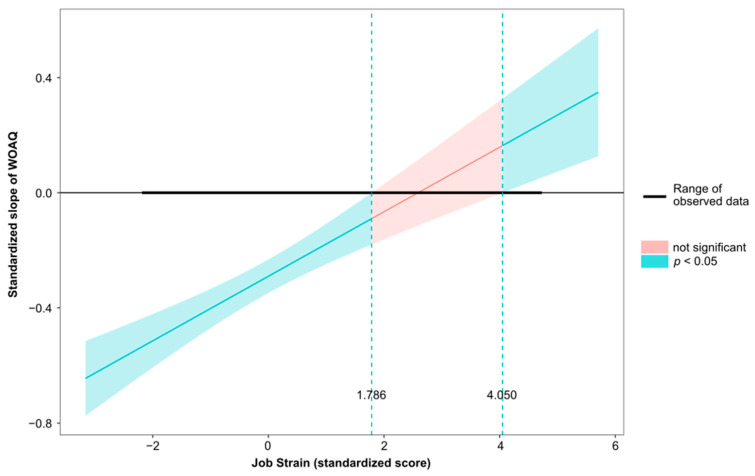
Johnson–Neyman Plot of the relationship between work organization quality and psychological health, moderated by job strain.

**Figure 6 ijerph-19-13458-f006:**
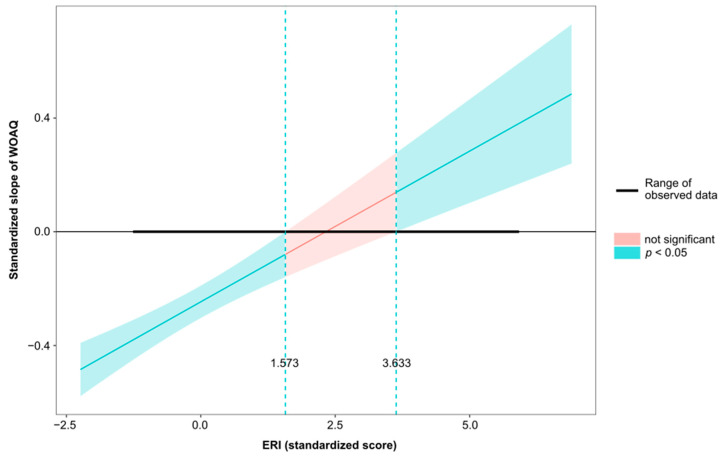
Johnson–Neyman Plot for of the relationship between work organization quality and psychological health, moderated by effort/reward imbalance.

**Table 1 ijerph-19-13458-t001:** Sample characteristics.

Sex	*n*	%
Male	115	33.3
Female	230	66.7
**Category**	** *n* **	**%**
Physician	68	19.7
Nurse	201	58.3
Support staff	76	22.0

**Table 2 ijerph-19-13458-t002:** Distribution of work organization assessment (WOAQ score, mean ± standard deviation), occupational stress (DCS and ERI) and psychological health (GHQ12) by occupational group, gender, and age group.

	**1-Physician** (*n* = 68)	**2-Nurse** (*n* = 201)	**3-Support Staff** (*n* = 76)	***p* Value ***
Work Organization	83.7 ± 21.6	83.6 ± 20.6	81.6 ± 21.7	0.769
Demand	13.5 ± 3.09	13.2 ± 3.02	13.6 ± 3.22	0.518
Control	17.8 ± 3.23	16.6 ± 2.77	16.3 ± 3.36	0.006 1 vs. 2:0.015 1 vs. 3:0.009
Support	19.8 ± 3.86	19.9 ± 3.29	19.3 ± 3.97	0.430
Effort	15.7 ± 5.67	15.0 ± 5.05	13.4 ± 4.35	0.014 1 vs. 3:0.015
Reward	44.1 ± 9.29	42.6 ± 7.59	43.0 ± 9.53	0.440
Overcommitment	11.6 ± 5.41	11.8 ± 5.14	11.1 ± 4.52	0.527
Psychological Health	23.2 ± 5.22	23.9 ± 6.14	22.1 ± 5.17	0.071
	**Younger (<41 years)** (*n* = 133)	**Middle (41–50 years)** (*n* = 126)	**Older (>50)** (*n* = 86)	***p* Value ***
Work Organization	83.7 ± 19.85	84.3 ± 21.83	80.7 ± 21.63	0.450
Demand	13.5 ± 3.22	12.9 ± 2.80	13.6 ± 3.20	0.138
Control	16.6 ± 3.11	16.7 ± 2.84	16.8 ± 3.21	0.612
Support	19.5 ± 3.73	20.2 ± 3.30	19.4 ± 3.65	0.226
Effort	14.9 ± 5.40	14.1 ± 4.38	15.5 ± 5.58	0.153
Reward	42.8 ± 8.13	43.0 ± 8.75	43.3 ± 8.35	0.936
Overcommitment	11.5 ± 5.04	11.8 ± 5.16	11.5 ± 5.00	0.909
Psychological Health	22.9 ± 5.75	23.6 ± 5.83	23.8 ± 5.84	0.447
	**Male** (*n* = 115)	**Female** (*n* = 230)		***p* Value ****
Work Organization	86.2 ± 22.53	81.7 ± 20.12		0.061
Demand	13.2 ± 3.02	13.4 ± 3.11		0.562
Control	17.1 ± 3.34	16.6 ± 2.86		0.153
Support	19.6 ± 3.88	19.8 ± 3.40		0.685
Effort	14.1 ± 4.85	15.1 ± 5.18		0.095
Reward	44.0 ± 8.99	42.5 ± 8.05		0.111
Overcommitment	10.8 ± 4.92	12.0 ± 5.09		0.029
Psychological Health	22.0 ± 4.32	24.1 ± 6.30		0.002

Note: (*) One-way ANOVA with Bonferroni post-hoc test (**) Student’s *t*-test.

**Table 3 ijerph-19-13458-t003:** Correlations between work organization quality, job strain, effort/reward imbalance, support, overcommitment, and psychological health. Values in the lower triangle are zero-order correlations; values in the upper triangle are partial correlations after controlling for gender and age.

	1	2	3	4	5	6
1. Work Organization	1	−0.557 ***	0.546 ***	−0.563 ***	−0.436 ***	−0.421 ***
2. Job Strain	−0.565 ***	1	−0.493 ***	0.579 ***	0.370 ***	0.245 ***
3. Support	0.551 ***	−0.496 ***	1	−0.510 ***	−0.286 ***	−0.165 **
4. ERI	−0.577 ***	0.570 ***	−0.525 ***	1	0.570 ***	0.421 ***
5. Overcommitment	−0.459 ***	0.377 ***	−0.297 ***	0.584 ***	1	0.553
6. Psychological health	−0.436 ***	0.245 ***	−0.160 **	0.423 ***	0.560 ***	1

Note: ***: *p* < 0.001; **: *p* < 0.01.

**Table 4 ijerph-19-13458-t004:** Adjusted direct and indirect associations of work organization with psychological health (measured by GHQ12) moderated via job strain. Moderation analysis.

Measure	Unstandardized Coefficients	Standard Error	*t*	*p* Value	Standardized Coefficients	η^2^
WOAQ	–0.081 (–0.113, –0.049)	0.016	–4.973	<0.001	−0.290 (−0.405, −0.175)	0.240
Job Strain	–0.064 (–2.179, 2.052)	1.075	–0.059	0.953	−0.003 (−0.113, 0.106)	0.000
Interaction WOAQ × Job strain	0.104 (0.036, 0.172)	0.035	3.009	0.003	0.136 (0.047, 0.225)	0.027
Sex (Female)	1.143 (0.054, 2.233)	0.554	2.065	0.040	0.093 (0.004, 0.181)	0.012
Age	0.067 (0.009, 0.125)	0.030	2.255	0.025	0.100 (0.013, 0.187)	0.014
Support	0.163 (–0.010, 0.335)	0.088	1.857	0.064	0.099 (−0.006, 0.204)	0.015
Overcommitment	0.537 (0.426, 0.648)	0.057	9.512	<0.001	0.463 (0.367, 0.559)	0.225
R^2^	0.403					

Note: Adjusted for age, sex, support and overcommitment; η^2^ values can be interpreted as negligible when η^2^ < 0.01, as small when 0.01 ≤ η^2^ < 0.06, as moderate when 0.06 ≤ η^2^ < 0.14, and large when η^2^ ≥ 0.14 [63].

**Table 5 ijerph-19-13458-t005:** Adjusted direct and indirect associations of work organization with psychological health (measured by GHQ12) moderated via effort/reward imbalance. Moderated linear regression analysis.

Measure	Unstandardized Coefficients	Standard Error	*t*	*p* Value	Standardized Coefficients	η^2^
WOAQ	−0.069 (−0.100, −0.038)	0.016	–4.338	<0.001	−0.247 (−0.359, −0.135)	0.241
ERI	3.609 (1.449, 5.768)	1.098	3.288	0.001	0.237 (0.095, 0.380)	0.073
Interaction WOAQ × ERI	0.077 (0.028, 0.127)	0.025	3.102	0.002	0.169 (0.062, 0.276)	0.029
Sex (Female)	1.231 (0.148, 2.315)	0.551	2.236	0.026	0.100 (0.012, 0.188)	0.012
Age	0.064 (0.006, 0.122)	0.029	2.179	0.030	0.096 (0.009, 0.182)	0.015
Support	0.273 (0.100, 0.447)	0.088	3.097	0.002	0.166 (0.061, 0.272)	0.042
Overcommitment	0.459 (0.337, 0.581)	0.062	7.391	<0.001	0.396 (0.290, 0.501)	0.163
R^2^	0.409					

Note: Adjusted for age, sex, support, and overcommitment; η^2^ values can be interpreted as negligible when η^2^ < 0.01, as small when 0.01 ≤ η^2^ < 0.06, as moderate when 0.06 ≤ η^2^ < 0.14, and as large when η^2^ ≥ 0.14 [63].

**Table 6 ijerph-19-13458-t006:** Relationship between work organization, occupational stress, and the risk of having taken sickness absence for musculoskeletal disorders.

Variable	Model I (Unadjusted) OR (95% CI)	*p* Value	Model II (Adjusted) OR (95%CI)	*p* Value	Nagelkerke R^2^
Work Organization	0.98 (0.97, 0.99)	0.003	0.98 (0.97, 0.99)	0.009	0.120
Job Strain	1.72 (0.79, 3.71)	0.169	1.82 (0.81, 4.11)	0.149	0.102
Social Support	0.96 (0.90, 1.03)	0.246	0.96 (0.89, 1.02)	0.198	0.100
ERI	1.63 (0.91, 2.93)	0.103	1.61 (0.87, 2.97)	0.127	0.102
Overcommitment	1.05 (1.00, 1.10)	0.048	1.94 (0.99, 1.09)	0.087	0.105

Note: Model II was adjusted for age, sex, and job type.

**Table 7 ijerph-19-13458-t007:** Relationship between work organization, occupational stress, and the risk of having taken sickness absence for disorders other than musculoskeletal problems.

Variable	Model I (Unadjusted) OR (95% CI)	*p* Value	Model II (Adjusted) OR (95%CI)	*p* Value	Nagelkerke R^2^
Work Organization	0.98 (0.97, 0.99)	<0.001	0.98 (0.97, 0.99)	<0.001	0.074
Job Strain	1.50 (0.73, 3.08)	0.096	1.51 (0.73, 3.11)	0.269	0.012
Social Support	0.89 (0.84, 0.95)	0.001	0.89 (0.84, 0.95)	0.001	0.057
ERI	2.00 (1.10, 3.63)	0.023	2.01 (1.10, 3.67)	0.023	0.028
Overcommitment	1.02 (0.97, 1,06)	0.455	1.01 (0.97, 1.06)	0.521	0.008

Note: Model II was adjusted for age, sex, and job type.

## Data Availability

The data are freely available on the Zenodo repository: https://zenodo.org/badge/DOI/10.5281/zenodo.7070452.svg accessed on 12 September 2022.

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
