# Peer review of "The Impact of Quality of Work Organization on Distress and Absenteeism among Healthcare Workers"

_ijerph, 2022, doi:10.3390/ijerph192013458_

Round 1

Reviewer 1 Report

quite interesting work that analyzes how the quality of work organization can be responsible not only for reduced productivity but also for an increased risk of mental and physical disorders.

requires a review of the structure of materials and methods, and possible updating of the bibliography, and in the conclusions, some suggestions on possible improvements.

Author Response

Reviewer #1

quite interesting work that analyzes how the quality of work organization can be responsible not only for reduced productivity but also for an increased risk of mental and physical disorders.

requires a review of the structure of materials and methods, and possible updating of the bibliography, and in the conclusions, some suggestions on possible improvements.

Thank you for giving me this opportunity to review the manuscript.

The manuscript submitted for publication by Magnavita et al., titled: " The Impact of Quality of Work Organization on Distress and 2 Absenteeism among Healthcare Workers ", is a paper that explores the influence of quality of work organization on work productivity. In my opinion, the manuscript is written quite well. The results are well analyzed and described proficiently. I just have a few suggestions for authors to make the manuscript more complete.

as regards the English language, I have not the skills to be able to assess the possible need for a revision of its form.

Response: We thank the reviewer for the appreciation expressed towards our work and for taking the time to review it.

ABSTRACT

Authors should state the aim of the work.

Response: The synthetic nature of the abstract prevents the study hypotheses from being fully reported; however, we had indicated the purpose of the work in the first line of the abstract. Following the indication of the reviewer we have added an explanatory sentence: “This study was aimed at testing this hypothesis.”

INTRODUCTION

Line 64. In addition to environmental factors, the authors could also briefly discuss the influence of epigenetic factors on job stress.

DOI -10.3390/ijerph16204031

Cannizzaro, E.; Ramaci, T.; Cirrincione, L.; Plescia, F. Work-Related Stress, Physio-Pathological Mechanisms, and the Influence of Environmental Genetic Factors. Int. J. Environ. Res. Public Health 201916, 4031. https://doi.org/10.3390/ijerph16204031

R.: We have read and enjoyed the study conducted by the reviewer and his collaborators. We referenced the relationship between genetic and epigenetic factors in the stress response and cited this study.

MATERIALS AND METHODS

Please rearrange this part by indicating key missing data such as: the number, gender, membership in the identified homogeneous groups, the inclusion and exclusion criteria. The description of the tests used, should be divided into paragraphs.

Response: We have added more information on the characteristics of the sample. We gladly accepted the advice to divide the section into subsections.

RESULTS

Line 209 to 215, to be included in the materials and methods paragraph along with the table.

Response: According to the STROBE checklist, Participants #13, reporting numbers of individuals at each stage of study—eg numbers potentially eligible, examined for eligibility, confirmed eligible, included in the study, completing follow-up, and analysed, should be placed in the Results section. The same is for #14, Descriptive data (characteristics of study participants and information on exposures and potential confounders).

DISCUSSION

Does the quality of the work environment influence the use of substances of abuse? Briefly discuss. https://doi.org/10.3390/ijerph182413196

Plescia, F.; Cirrincione, L.; Martorana, D.; Ledda, C.; Rapisarda, V.; Castelli, V.; Martines, F.; Vinnikov, D.; Cannizzaro, E. Alcohol Abuse and Insomnia Disorder: Focus on a Group of Night and Day Workers. Int. J. Environ. Res. Public Health 202118, 13196. https://doi.org/10.3390/ijerph182413196

Response: We enjoyed the study on alcohol abuse and insomnia, but we believe it is beyond the scope of this work.

Reviewer 2 Report

The manuscript presents an interesting purpose, aiming at evaluate 4 different hypotheses concerning the relatioship between the work organisation, stress and health outcomes. The text is well written and the content is relevant. The background is consistent and the hypotheses are clear. A few clarifications concerning the methods and few revisions in results presentation may be useful for readers. Comments are made below for each section.

The introduction described adequately the state-of-art of literature and the context of health labour in Italy. Previous literature cited by the authors is relevant and strongly justify the study. Please revise a typo in line 77 (page 2) ("sorganisation's"). Also a revision in the sentence "This is especially pronounced in emotional labour which common in healthcare (line 89) may be necessary). The sentence in lines 91-92 are also not clear to me ("which [emotion] can be conceptualised as perceptions of stress or strain and measured using two complementary dimensions, job strain [35] and effort-reward imbalance [36]." Is it correct that "emotion" can be "conceptualised" as perceptions of stress and measured by job strain and ERI models? Both models measure occupational stress and have a strong relationship with the work organisation, not only with individuals "emotions". It might be interesting to clarify this aspect in the introduction.

In the introduction, the authors also cited that examining relationships between the work organisation and health, "it is important to control for overcommitment and social support because these factors are also strong independent predictors of psychological health." Well, the fact that overcommitment and social support predict psychological health by itself is not a strong justification to control in the analyses, once other assumptions should be fit (being associated with the exposure and not in the causal path between the main exposure and the outcome). Is it not possible that those characteristics are mediators of this association?

Methods: Is the study a census of a local health unit? Where is it located? This health unit is responsible for which area/population? Regarding the measurements, several validated instruments are described. The literature describes different ways of categorising scores of job strain and ERI, defining different levels/categories of stress, for example. However, no categorisations of measures are specified by the authors. Did the authors considered all measures as continuous variables? 

Results: Table 1 shows very few characteristics of the sample (only sex and profession). It would be interesting the see the description of frequencies of other characteristics, such as age, ethnicity, income, marital status, etc). Why this was not presented? 

Table 2: Did the authors compared means? ANOVA and t test were used? Were data regarding psychosocial aspects of work normally distributed to assure a adequate use for those statistical tests?

Table 3: Psychosocial aspects of work are highly correlated... Are they not measuring the "same thing"? What differs between them? This might be discussed further in the next section.

Tables 6 and 7: When showing ORs, I would suggest to use only two decimals after dot. 

The fact that the authors seem to have used only continuous variables for most analyses, the magnitude of effects may be more difficult to interpret. This might be further discussion in the discussion section

Concerning the main aspects of the study subject, the tested hypotheses and the presented results, the discussion is interesting and raise relevant points to the topic. A few additional aspects may be addressed in light of adjustments that may be done in the analyses, if relevant to authors. 

Author Response

Reviewer #2

The manuscript presents an interesting purpose, aiming at evaluate 4 different hypotheses concerning the relatioship between the work organisation, stress and health outcomes. The text is well written and the content is relevant. The background is consistent and the hypotheses are clear. A few clarifications concerning the methods and few revisions in results presentation may be useful for readers. Comments are made below for each section.

The introduction described adequately the state-of-art of literature and the context of health labour in Italy. Previous literature cited by the authors is relevant and strongly justify the study. Please revise a typo in line 77 (page 2) ("sorganisation's"). Also a revision in the sentence "This is especially pronounced in emotional labour which common in healthcare (line 89) may be necessary). The sentence in lines 91-92 are also not clear to me ("which [emotion] can be conceptualised as perceptions of stress or strain and measured using two complementary dimensions, job strain [35] and effort-reward imbalance [36]." Is it correct that "emotion" can be "conceptualised" as perceptions of stress and measured by job strain and ERI models? Both models measure occupational stress and have a strong relationship with the work organisation, not only with individuals "emotions". It might be interesting to clarify this aspect in the introduction.

Response: We sincerely thank the reviewer for the appreciation he/she expressed on our work. We promptly corrected the typo he reported to us, as well as other typographical errors that appeared in the review of the text by the editorial secretariat.

A complete discussion of the concept of emotion work  or emotional labor is obviously beyond the scope of this article. Interested readers will find a satisfactory discussion in the cited reference. We have added an explanation of the term and we corrected the sentence of lines 91-92 which, due to excessive synthesis, was unclear; we have added a short explanation. The text now is as follows: …

In the introduction, the authors also cited that examining relationships between the work organisation and health, "it is important to control for overcommitment and social support because these factors are also strong independent predictors of psychological health." Well, the fact that overcommitment and social support predict psychological health by itself is not a strong justification to control in the analyses, once other assumptions should be fit (being associated with the exposure and not in the causal path between the main exposure and the outcome). Is it not possible that those characteristics are mediators of this association?

Response: We thank the reviewer for the remark on the term "control for" which we had inappropriately used in the introduction. In this first part of the article we should have simply said that social support and overcommitment must be taken into account; we have modified the text in this way.

In the next methodological part of the paper, we explain that we have used these variables as confounding factors, because their role is not of mediators as they are not directly produced by the factor under study which is the organization of work.

Methods: Is the study a census of a local health unit? Where is it located? This health unit is responsible for which area/population? Regarding the measurements, several validated instruments are described. The literature describes different ways of categorising scores of job strain and ERI, defining different levels/categories of stress, for example. However, no categorisations of measures are specified by the authors. Did the authors considered all measures as continuous variables? 

R.: We have added the required population information. In this study, the variables expressing the quality of organization, stress and mental health were treated as continuous

Results: Table 1 shows very few characteristics of the sample (only sex and profession). It would be interesting the see the description of frequencies of other characteristics, such as age, ethnicity, income, marital status, etc). Why this was not presented? 

Response: The age is reported in the text immediately before Table 1. Ethnicity was homogenous, the sample was entirely composed of Italians, whites. The income of health workers is determined in Italy by the national employment contract which establishes the salary of the various categories. Family status was not recorded in this survey.

Table 2: Did the authors compared means? ANOVA and t test were used? Were data regarding psychosocial aspects of work normally distributed to assure a adequate use for those statistical tests?

R.: As reported in the footnote of Table 2, mean comparisons for occupation and age group were carried out using ANOVA and Bonferroni-corrected post-hoc tests, while sex comparisons were carried out using independent-sample t-tests. As reported by Lumley et al. (2002), the assumption of normality is only required for small samples, due to the central limit theorem. With sample sizes exceeding 30, as it is the case here, violations of the normality assumptions are not problematic, and they become less and less problematic, even when extreme, as the sample size increases.

Lumley, T., Diehr, P., Emerson, S., & Chen, L. (2002). The importance of the normality assumption in large public health data sets. Annual Review of Public Health, 23(1), 151–169. https://doi.org/10.1146/annurev.publhealth.23.100901.140546

Table 3: Psychosocial aspects of work are highly correlated... Are they not measuring the "same thing"? What differs between them? This might be discussed further in the next section.

R.: Although the pattern of correlation of Job Strain, Support, ERI, and Overcommitment suggest so (and an exploratory factor analysis carried out on these variables revealed that a single overarching factor accounts for 49% of variance), it should be considered that the pattern of correlations of these variables with Psychological Health is not homogeneous. When we tested whether the null hypothesis that in the population the (absolute) correlations of Psychological Health with the four variables mentioned earlier were equal, we found that this was not the case, with a moderate/strong effect size (2(3) = 85.503, p < .001, r = .492). Post-hoc tests revealed that the correlation of Psychological Health with any psychosocial variable was statistically different from the correlation of Psychological Health with any other, except for Job Strain vs Support. This means that, although psychological variables can be considered as facets of a common variable, they do not equally predict psychological health outcomes.

Tables 6 and 7: When showing ORs, I would suggest to use only two decimals after dot. 

Response: We have accepted the invitation and we have removed a figure from the numbers displayed

The fact that the authors seem to have used only continuous variables for most analyses, the magnitude of effects may be more difficult to interpret. This might be further discussion in the discussion section

R.: We added a measure of effect size (eta-square) to Table 4 and Table 5

Concerning the main aspects of the study subject, the tested hypotheses and the presented results, the discussion is interesting and raise relevant points to the topic. A few additional aspects may be addressed in light of adjustments that may be done in the analyses, if relevant to authors. 

R.: We thank the reviewer very much for the attention with which he/she reviewed this study and we hope that the improved version by accepting his/her suggestions is now acceptable.